# Synthesis, Characterization, and Evaluation of Antimicrobial Efficacy of Reduced Graphene–ZnO–Copper Nanocomplex

**DOI:** 10.3390/antibiotics12020246

**Published:** 2023-01-25

**Authors:** Varish Ahmad, Mohammad Omaish Ansari

**Affiliations:** 1Health Information Technology Department, The Applied College, King Abdulaziz University, Jeddah 21589, Saudi Arabia; 2Centre of Artificial Intelligence for Precision Medicines, King Abdulaziz University, Jeddah 21589, Saudi Arabia; 3Center of Nanotechnology, King Abdulaziz University, Jeddah 21589, Saudi Arabia

**Keywords:** nanocomposite, reduced graphene oxide, antibacterial, drug resistant, pathogen

## Abstract

The prevalence of antibiotic-resistant diseases drives a constant hunt for new substitutes. Metal-containing inorganic nanoparticles have broad-spectrum antimicrobial potential to kill Gram-negative and Gram-positive bacteria. In this investigation, reduced graphene oxide-coated zinc oxide–copper (rGO@ZnO–Cu) nanocomposite was prepared by anchoring Cu over ZnO nanorods followed by coating with graphene oxide (GO) and subsequent reduction of GO to rGO. The synthesized nanocomposite was characterized by scanning electron microscopy, transmission electron microscopy, elemental analysis, and elemental mapping. Morphologically, ZnO–Cu showed big, irregular rods, rectangular and spherical-shaped ZnO, and anchored clusters of aggregated Cu particles. The Cu aggregates are spread uniformly throughout the network. Most of the ZnO particles were partially covered with Cu aggregates, while some of the ZnO was fully covered with Cu. In the case of rGO@ZnO–Cu, a few layered rGO sheets were observed on the surface as well as deeply embedded inside the network of ZnO–Cu. The rGO@ZnO–Cu complex exhibited antimicrobial activity against Gram-positive and Gram-negative bacteria; however, it was more effective on *Staphylococcus aureus* than *Escherichia coli*. Thus, rGO@ZnO–Cu nanocomposites could be an effective alternative against Gram-positive and Gram-negative bacterial pathogens.

## 1. Introduction

Antibiotic-resistant pathogens are increasingly common, which has prompted an ongoing quest for novel substitutes. Food-borne and water-borne bacterial species are one group of drug-resistant pathogens that seriously endanger public health because they cause disease outbreaks [1].Various metal nanoparticles are harmful to a number of drug-resistant, Gram-negative, and Gram-positive disease-causing microbial species [2,3,4]. Although inorganic nanoparticles are well known for their wide range of biocidal properties, their potent antimicrobial effectiveness mechanism is less well characterized. However, reactive oxygen species, which are hazardous to bacteria, are believed to be produced when ions are released into solution. Other research has demonstrated that nanoparticles can kill cells by penetrating bacterial cell walls and attacking organelles.

In contrast to organic antibiotics, their inorganic equivalents fight drug-resistant bacteria through many routes [5]. Nanocomplexes of metal oxides are highly regarded for their bacteriostatic and bactericidal properties [6,7]. Metal oxides are easily synthesized, and the shape and size of the nanocomplex can be modified cost-effectively. Although the significant antimicrobial potential of silver and gold nanomaterials has been explored, the resulting compounds are unaffordable due to their high cost. However, copper oxide and zinc oxide have demonstrated different antibacterial activities against many bacterial and fungal pathogen species [8]. Due to their potential antibacterial properties, they have been employed for killing phytopathogens, food pathogens, and infectious pathogens. However, Gram-positive and Gram-negative bacteria have not yet been used to test the antibacterial properties of these nanomaterials.

ZnO nanoparticles have received a large amount of attention in recent years due to their potential use as an antimicrobial, a photocatalyst, or an additive to industrial products. Moreover, the effectiveness and functionality of ZnO nanoparticles and nanocomplexes can be improved by extending or changing their surface area via complexation with dopants from other nanoscale materials, such as Mn, Cu, and Fe [9]. This work investigates the antibacterial activity of copper oxide, zinc oxide, and rGO@ZnO–Cu nanocomposites for Gram-positive and Gram-negative microorganisms. The inquiry studied the antibacterial activity of each nanocomposite against the Gram-positive and Gram-negative bacteria species, *Escherichia coli* and *Staphylococcus aureus*.

The graphene family has excellent surface qualities and the sheets have strong inter-planar interactions which makes them insoluble due to the aggregation of particles thus limiting their antimicrobial properties. Thus, reducing aggregation using nanocarbon–polymer-based hybrid dispersion systems can increase solubility, stability, and antimicrobial action [10]. Graphene–metal metal oxide composites and graphene–polymer composites were tested against many bacterial indicators, including *Pseudomonas aeruginosa*, *E. coli*, *P. syringae*, *Xanthomonas campestris* pv. *campestris*, *Enterococcus faecalis, Salmonella typhimurium, Streptococcusmutans*, *Fuscobacteriumnucleatum*, and *Porphyromonasgingivalis*. Reduced graphene oxide (rGO), as well as other graphene-based conjugated nanomaterials, have been reported to be effective against *S. aureus*, *P.aeruginosa*, *S.typhimurium*, *E. coli*, and *E.faecalis*. The inhibitory effects observed from the conjugated form were greater than those observed with the detached form due to an additive effect. The synergistic antibacterial effect of GO–ZnO with low cytotoxicity has also been reported [11]. The GO–PVK conjugated nanoform significantly inhibited the biofilm formation in *Rhodococcusopacus*, *E. coli*, *Cupriavidus metallidurans,* and *Bacillus subtilis* [12].

Although silver and gold nanoparticles have low dispersibility, high antimicrobial potency, low biocompatibility, and high toxicity. The graphene oxide and other metal oxide-based nanomaterials, including Cu and Zn, are intriguing prospects for many biomedical applications [13,14]. In this study, zinc oxide and copper oxide nanoparticles were complexed with graphene oxide, and the resulting nanocomposites were characterized by scanning electron microscopy (SEM), transmission electron microscopy (TEM), antibacterial activity, and computational interaction analysis. The synergies between the three nanoparticles were applied to increase the antibacterial activity of the nanocomposites.

## 2. Results

### 2.1. Synthesis and Characterization of rGO@ZnO–Cu

The ZnO–CuandrGO@ZnO–Cu nanomaterials were prepared by the chemical method. The steps involved in the synthesis of rGO@ZnO–Cu are shown in Figure 1.

The SEM micrographs of the ZnO–Cu and rGO@ZnO–Cu nanocomposites are presented in Figure 2. Morphologically, ZnO–Cu has large irregular rods composed of rectangular and spherical ZnO coated with clusters of aggregated Cu particles. The Cu aggerates are uniformly spread throughout the network, and most of the ZnO particles are partially covered with Cu aggregates, while some of the ZnO is fully covered with Cu (Figure 2a–c). In the case of rGO@ZnO–Cu, a few layered rGO sheets can be seen on the surface, as well as deeply embedded inside the network of ZnO–Cu (Figure 2d–f). The interconnected rGO sheets provide a network to which ZnO–Cu attaches. The TEM images show that a few layers of rGO are well connected with ZnO and Cu. The size of the ZnO is in the range of 500 nm, while that of Cu is in the range of ~50–150 nm (Figure 3).

The presence of all the included elements, i.e., C, O, Cu, and Zn, in both elemental maps and EDS data, as well as their uniform distribution, suggests the efficacy of the synthesis methodology (Figure 4). The EDS represents the presence of specific elements (C, O, Cu, and Zn), while the maps give their distributions.

### 2.2. Antibacterial Activity

The antimicrobial activity of the nanocomposites Cu, ZnO, and rGO@ZnO–Cu was determined on agar plates using the agar well diffusion assay (Table 1), where the antimicrobial activity was seen in the form of an empty zone around the well of agar media (Figure 5).The maximum antibacterial activity was observed with rGO@ZnO–Cu, followed by ZnO and Cu. The inhibitory effect of rGO@ZnO–Cu was stronger against Gram-positive bacteria than Gram-negative bacteria. Moreover, the dose-dependent and synergistic growth inhibitory effects were observed and increased with complexation (Cu toZnO–Cu and rGO@ZnO–Cu: ZOI;11–13–17mm).The effects of rGO@ZnO–Cu were stronger than those of ZnO–Cu (Figure 4)**.** The MIC for the antimicrobial activity of rGO@ZnO–Cu against Gram-positive *S. aureus* was 35 µg/mL, and for Gram-negative *E. coli*, it was 45 µg/mL. The cells of the tested bacteria survived under the concentrations, and the survival rate in ZnO–Cu was higher than in rGO@ZnO–Cu. Antibacterial activities of nanoparticles against *E. coli* and *S. aureus* were reported. The antibacterial effects are shown as a zone of inhibition in millimeters in Figure 5A,B.

### 2.3. The Proposed Antibacterial Mechanism of rGO@ZnO–Cu

The properties of rGO@ZnO–Cu could promote microbial cell death, as can be seen in Figure 5.TherGO@ZnO–Cu nanoparticles are deposited on the surface of the bacteria and subsequently cause death as the bacteria lysed. The synthesized rGO@ZnO–Cu potentially interacts strongly with Gram-positive and Gram-negative bacteria, but the inhibitory effect was stronger with Gram-positive bacteria. This difference could be due to differences in the interactions of the nanoparticle with the cellular membrane of each tested bacteria. The studied bacteria may die as a result of cell lysis caused by the nanoparticles being deposited on the surface of the microbial cells. (Figure 6).

The docking model demonstrating the interaction of the synthesized nanoparticles with the glycan in the bacterial membrane is shown in Figure 7a and Table 1, whereas the ZnO–Cu–glycan interaction is shown in Figure 7b. The bacterial glycan binds to both nanoparticles, as seen in the SEM image of the bacterial cells treated with rGO@ZnO–Cu (Figure 6). The results of the docking study favored rGO–glycan, followed by Cu–glycan and ZnO–glycan. In the interaction of rGO with the bacterial target, a hydrogen bond (:LIG2:N—A:GLU83:O) with a bond length of 2.23976 Å was also observed. As a result, computational analysis revealed the produced nanoparticles to be a substantial antibacterial system, and their synthesis and antimicrobial significance are presented in Figure 8.

## 3. Discussion

The results confirm the successful synthesis of rGO@ZnO–Cu, ZnO–Cu, and ZnO. The parameters of the synthesis process and the type of approach affect the surface properties of nanomaterials and vary for different methods [15]. The use of nanoparticles as treatments, diagnostic sensors, and vehicles for the transport of nanoforms into the cell is influenced by their surface qualities, such as area, functional groups, hydrophilicity, and hydrophobicity [16].

Our study synthesized a nano system with variable shapes, such as irregular rods, rectangular ZnO–Cu, and spherical ZnO, with a coating of clusters of aggregated Cu particles. Moreover, the observed size was significant for its therapeutic use and antimicrobial action. The presence of variable shapes and sizes in the produced nano system indicated that this system could benefit human welfare for many applications, including sensing, antimicrobials, and drug delivery [13,14]. Many researchers claim that varied nanoparticles exhibited noticeably antimicrobial effect, higher margination and deposition rates, including metal particles and liposomes with varying sizes (60–130 nm) and forms (spherical, rod). [17,18]. Moreover, in the system, the Cu aggregates were uniformly spread throughout the network, and most of the ZnO particles were partially covered with Cu aggregates, while some of the ZnO particles were completely covered with Cu [19,20].

Antibiotic resistance is a common problem, which drives a continuous search for novel treatments. Gram-positive and Gram-negative bacteria can be killed by the broad-spectrum antibacterial capability of inorganic metal nanoparticles [20,21].The graphene possesses high specific surface area, biocompatibility, good diffraction strength, high Young’s modulus, rapid ion migration, and high electrical and thermal conductivities. Although carbon nanomaterials, particularly those from the graphene family, have great surface properties, they also have strong inter-planar interactions that limit their antibacterial activities and render them insoluble due to particle aggregation. As a result, utilizing hybrid dispersion systems based on nanocarbon polymers has been shown to boost solubility, stability, and antibacterial effects [22,23].

Thus, the antimicrobial potentials of the synthesized nano system were significant against both Gram-positive and Gram-negative tested bacteria, and the inhibitory effect increased with the dose and combination of metals ion the order rGO@ZnO–Cu, ZnO–Cu, and Cu. It has previously been shown that a greater inhibitory zone was produced by nanomaterials with higher concentrations of ZnO NPs. Additionally, most samples showed a higher antibacterial zone of inhibition against *E. coli* than *S. aureus*, indicating that Gram-negative microorganisms have a higher sensitivity to the synthesized nanoparticles [24].

However, due to the presence of peptidoglycan, Gram-negative bacteria have a more complex cell wall than Gram-positive bacteria, which could result in weaker interactions between the functional groups of the cell wall of Gram-negative bacteria and nanomaterials. However, the doped Cu–ZnO particles had stronger inhibitory effects on Gram-positive bacteria than on Gram-negative bacteria. Moreover, dose-dependent synergistic effects have also been observed with this nano system [25].

*E. coli* has also been shown to be resistant to the synergistic antibacterial activity of GO–ZnO with little cytotoxicity. To overcome this problem, rGO@ZnO–Cu was synthesized, which showed greater antibacterial activity against the tested bacteria. This could be due to the synergistic antimicrobial effect mediated by Cu, ZnO, and rGO. The substantial bactericidal activities of the rGO–ZnO hybrid nanomaterials coupled with the biodegradable polymer Poly (3-hydroxybutyrate-co-3-hydroxyvalerate), PHBV have been reported to increase the effectiveness [26]. In the case of rGO@ZnO–Cu, a few layered rGO sheets were seen on the surface, as well as deeply embedded inside the ZnO–Cu network, which could also be more effective against the tested bacteria.

The most common bacterial pathogen that causes skin and soft tissue infections is *S. aureus*. Uncertainties still exist regarding the defense mechanisms employed by the host immune cell against infection by the bacterial cells, which could be due to the involvement of glycan linkage. The rGO@ZnO–Cu nanoparticle could express its bactericidal action through the lysis of bacterial cells, as observed in the SEM image of bacterial cells treated with rGO@ZnO–Cu. This hypothesis was further supported by docking studies of the immunogenic glycan molecule found in bacterial cell walls, which mediated the stronger binding of rGO@ZnO–Cu nanoparticles to the surface of the bacterial cell, which could result in the lysis of the bacterial cell [18]. The surface properties of the synthesized nanoparticles favor greater interactions [13]. The interactions of synthesized nanoparticles were also supported by the docking analysis, in which rGO@ZnO–Cu interacted strongly with the cell membrane through the glycan in the bacterial cell wall. The binding energy of rGO@ZnO–Cu was more negative than those of ZnO–Cu and Cu. The lesser the binding or docking energy (more negative) of the ligand and receptor, the stronger the interaction is. Therefore, the rGO@ZnO–Cu nanocomplex had a stronger inhibitory effect against *S. aureus*. The rGO@ZnO–Cu nanocomplex was found to be a significant antimicrobial agent that can be used to control the growth of clinical and food-borne microbial pathogens.

## 4. Material and Methods

Analytical-grade reagents were used to synthesize Cu nanoparticles, ZnO nanotubes, rGO, and their composites. Copper sulfate pentahydrate (CuSO_4_·5H_2_O) was obtained from Fluka, and cetyl trimethyl ammonium bromide (CTAB) was obtained from Otto Chemicals. Zinc acetate, sulfuric acid, phosphoric acid, potassium permanganate, sodium hydroxide, and ascorbic acid (C_6_H_8_O_6_) were obtained from Sigma-Aldrich. Deionized water was used in the experiments.

### 4.1. Synthesis of rGO@ZnO–Cu Nanocomposites

To synthesize rGO@ZnO–Cu nanocomposites, ZnO nanorods were first anchored to Cu to obtain ZnO–Cu, which was then coated with GO and reduced. The synthesis of ZnO nanorods was achieved by the recrystallization process, as reported by Hossain et al. [12,27]. The synthesis of the GO stock solution used the method in our previous report [28]. To prepare ZnO–Cu, 100 mg of ZnO nanorods was placed in a 250 mL beaker, and 0.1 M copper (II) sulfate solution and 0.25 g of CTAB were added under stirring conditions. In another beaker, 50 mL of 0.2 M ascorbic acid solution was prepared. Finally, the solution of ascorbic acid was slowly added to the ZnO and copper (II) sulfate solution followed by the addition of 30 mL of 1 M sodium hydroxide solution. The whole system was heated to 80 °C for 2 h, and the resulting dark reddish-brown color confirmed the formation of Cu. The prepared ZnO–Cu was separated by centrifugation, washed with excess water and ethanol, and dried at room temperature [29]. To prepare rGO@ZnO–Cu, 1 mL of GO solution (5 mg/mL) was added to 150 mg of ZnO–Cu, and the resulting mixture was sealed in a crucible; a small hole was made in the crucible with a pin. The sealed ZnO–Cu and GO mixture was heated in an air furnace at 200 °C for 4 h to completely reduce the GO into rGO, to form rGO@ZnO–Cu.

### 4.2. Characterizations

Morphological analysis was conducted by field emission scanning electron microscopy (FESEM, JEOL, JSM-7600F, Tokyo, Japan) and transmission electron microscopy (HRTEM, JEOL, JSM, ARM-200F, Tokyo, Japan).The SEM images of Cu–ZnO, rGO@ZnO–Cu, and *S. aureus* treated with rGO@ZnO–Cu were recorded 10 kV [29]. To identify the elements present, energy-dispersive X-ray spectroscopy (EDS) analysis was performed using an Oxford energy-dispersive X-ray spectrometer, High Wycombe, UK), while elemental mapping studies were conducted to understand the distribution of the elements.

### 4.3. Antibacterial Activity

The antibacterial tests of each synthesized nanomaterial were conducted under strict sterile conditions using a laminar flow cabinet. The water, pipette, media, pipette tips, and glassware used to evaluate antibacterial activity were sterilized by autoclaving (UMB220 Benchtop Autoclave, Astell, Harrisburg, NC, USA) at 121 °C for 15 min. An agar well diffusion assay was used to evaluate the antimicrobial activity of the synthesized nanomaterials [28,29,30,31], and the antibacterial effects were measured using the zone of inhibition (millimeter) formed around the wells. All findings are presented as mean ± SEM. The data were analyzed using one-way ANOVA with Holm-Šídák’s multiple-comparisons test, which was performed using GraphPad Prism version 9.4.0 for MacOS, GraphPad Software, San Diego, CA, USA. In all cases, *p* < 0.05 was considered statistically significant. The reference bacterial strains *S. aureus* (ATCC6538) and *E. coli* (ATCC25922) were used to measure the antibacterial activity. The procedure involved preparing stock solutions of Cu, ZnO, and composites in sterile deionized water. The serial dilution method was used to create solutions of varying concentrations of the synthesized nanoparticles to determine antibacterial activities against *E. coli* and *S. aureus*, and MIC was analyzed by measuring the optical density at 590 nm [28].

### 4.4. Proposed Mechanism of rGO@ZnO–Cu

The surface morphology of *S. aureus* was analyzed by SEM. The treated bacteria exposed to 35 µg/mL of rGO@ZnO–Cu were incubated overnight at 37 °C, fixed, dehydrated with different concentrations of alcohol, and used for image analysis. The proposed antibacterial mechanism was described based on the effect observed in the SEM images and the docking interactions of each nanoparticle, which were evaluated with bacterial membrane proteins [28].

### 4.5. The Molecular Interaction (Docking Studies)

The docking was performed by preparing the ligand and receptor molecules, followed by molecular docking analysis. The 2D structural files of CuO (CID:14829), ZnO (CID:14806), and rGO (CID:297) and their SMILES IDs were downloaded from the PubChem Database (https://pubchem.ncbi.nlm.nih.gov/, accessed on 4 November 2022) [32]. NovoPro Lab server (https://www.novoprolabs.com/tools/smiles2pdb, accessed on 4 November 2022) was used to convert the SMILES IDs into 3D PDB files. Furthermore, the 3D crystal structure of the receptor peptidoglycan glycosyltransferase (PDB: 2OQO) was downloaded from Protein Data Bank (https://www.rcsb.org/structure/2oqo, accessed on 4 November 2022) [33,34]. The molecular interaction was performed over the bacterial peptidoglycan glycosyl transferase receptor. The interaction of CuO, ZnO, and rGO with peptidoglycan glycosyl transferase was modeled using the PatchDock online server (https://bioinfo3d.cs.tau.ac.il/PatchDock/, accessed on 4 November 2022).ThePatchDock tool uses a geometry-based molecular docking algorithm as a scoring function. Based on the ranking, the scores of the docked file were selected and subjected to the post-docking 3Dconformation analysis using Biovia Discovery Studio Visualizer 2021, which includes hydrogen bonds and length, and hydrophobic and other interactions, including most interacting amino acids [35,36].

## 5. Conclusions

This work employed a straightforward method to successfully synthesize a rGO@ZnO–Cu nanocomposite with a few layered rGO sheets on the surface, as well as deeply embedded inside the ZnO–Cu network. This study demonstrated that the synthesized rGO@ZnO–Cu nanocomposite was free of contaminants and had a high level of crystallinity. The synthesized nanoparticles exhibited good antimicrobial activities against Gram-negative and Gram-positive bacterial pathogens. Thus, based on their antibacterial action, the rGO@ZnO–Cu nanoparticles could be effective nano systems for a variety of industrial and domestic uses, including imaging, energy-based research, catalysis, environmental, and medical applications. Further research should be conducted in vitro and in vivo to further investigate the importance of rGO@ZnO–Cu to human welfare.

## Figures and Tables

**Figure 1 antibiotics-12-00246-f001:**
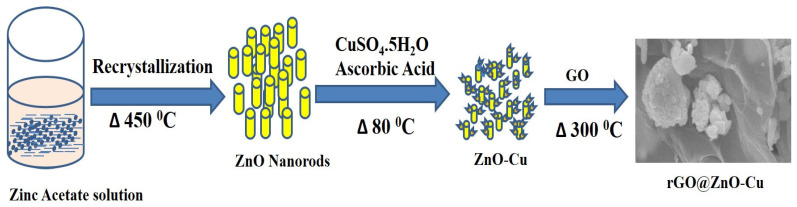
Schematic representation ofthe synthesis of rGO@ZnO–Cu nanocomposite.

**Figure 2 antibiotics-12-00246-f002:**
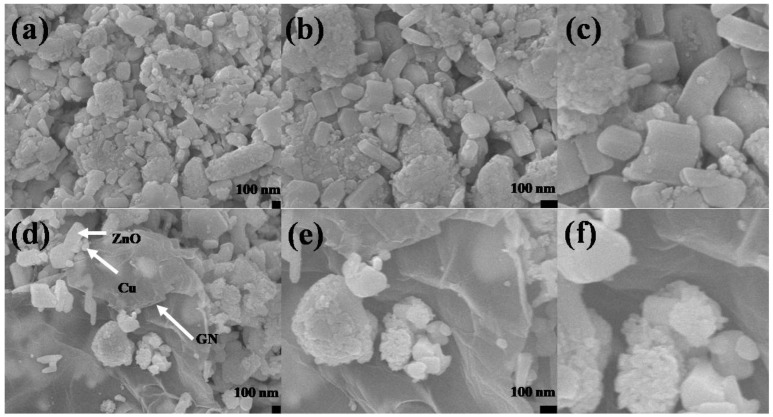
Scanning electron microscopy (SEM) images of (**a**–**c**) Cu–ZnO and (**d**–**f**) rGO@ZnO–Cu nanocomposites at different magnifications. (**a**,**d**) at 30,000×, (**b**,**e**) at 60,000×, and (**c**,**f**) at 120,000×.

**Figure 3 antibiotics-12-00246-f003:**
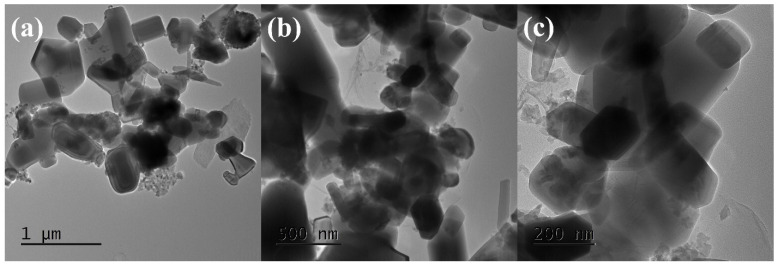
Transmission electron microscopy (TEM) images of rGO@ZnO–Cu nanocomposites at different magnifications. (**a**) 1 μm magnification; (**b**) 500 nm magnification and (**c**) 200 nm magni-fication.

**Figure 4 antibiotics-12-00246-f004:**
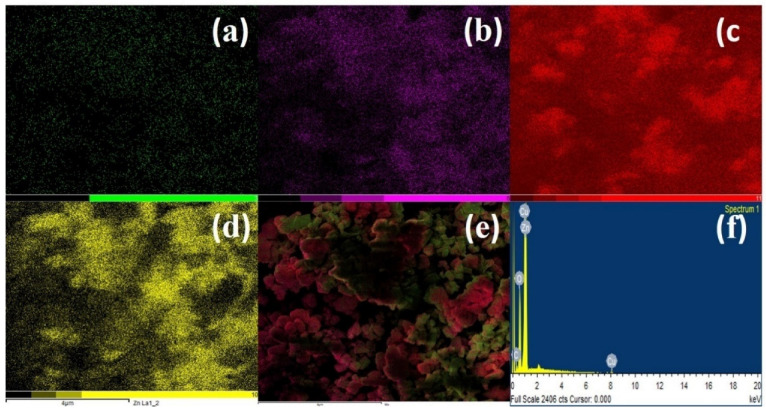
Elemental mapping of rGO@ZnO–Cu. (**a**) C, (**b**) O, (**c**) Cu, (**d**) Zn, and (**e**) all mapped elements, (**f**) EDAX analysis of rGO@ZnO–Cu.

**Figure 5 antibiotics-12-00246-f005:**
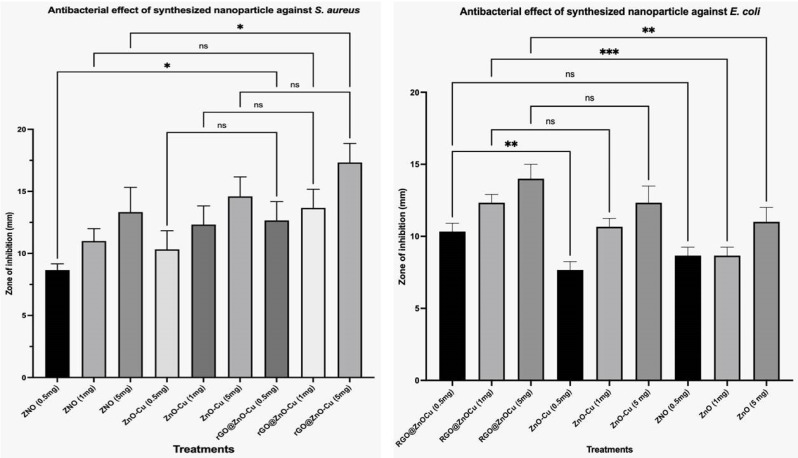
The antibacterial activity of rGO@ZnO–Cu nanoparticles against (**A**) *S. aureus* and (**B**) *E. coli.*, *p* < 0.05 was considered statistically significant. (* *p* < 0.05, ** *p* < 0.005, *** *p* < 0.001).

**Figure 6 antibiotics-12-00246-f006:**
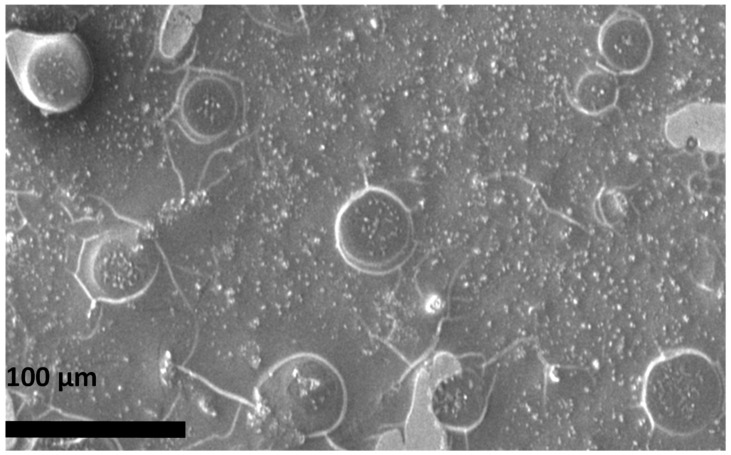
ESEM analysis of *S. aureus* treated with rGO@ZnO–Cu at 200× magnifications showing interaction with bacterial cells. Some cells have lysed due to deposition of rGO@ZnO–Cu on the surface.

**Figure 7 antibiotics-12-00246-f007:**
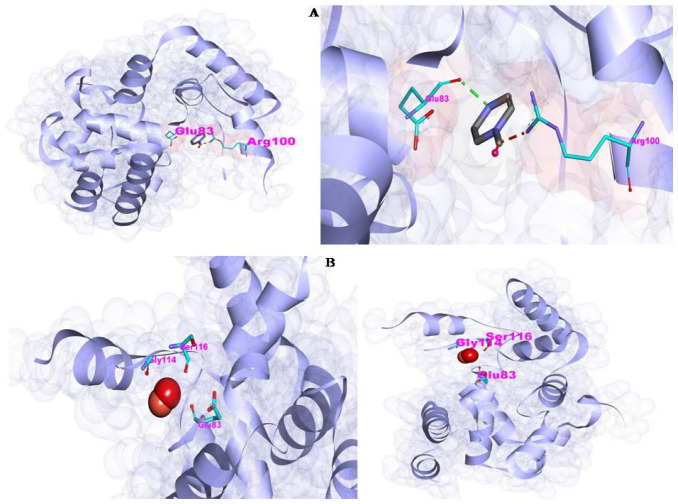
The binding interactions of synthesized rGO@ZnO–Cu nanoparticles with bacterial membrane glycan. In (**A**), the interaction with glycan is shown, and in (**B**), the interaction ofZnO–Cu with bacterial glycan is shown. Both synthesized particles bind with the bacterial glycan molecule through a hydrogen bond.

**Figure 8 antibiotics-12-00246-f008:**
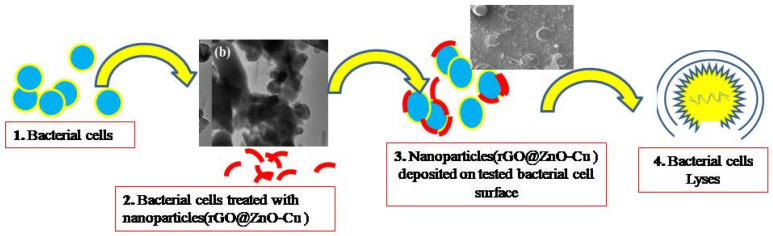
Conclusive steps from synthesis to interaction of rGO@ZnO–Cu with the *S. aureus* bacterial cells.

**Table 1 antibiotics-12-00246-t001:** Docking interaction analysis of each type of nanoparticle with the bacterial glycan molecules.

Complex	Atomic Contact Energy (ACE)	PatchDockScore	Hydrogen Bond	Hydrogen Bond Length
rGO–glycan	−61.36	2302	:LIG2:N - A:GLU83:O	2.23976
Cu–glycan	−12.53	1080	N/A	N/A
ZnO–glycan	2.71	660	:UNL1:O1 - A:ASN125:O	2.42469

## Data Availability

Not applicable.

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
