# Peer review of "Synthesis, Characterization, and Evaluation of Antimicrobial Efficacy of Reduced Graphene–ZnO–Copper Nanocomplex"

_antibiotics, 2023, doi:10.3390/antibiotics12020246_

Round 1
Reviewer 1 Report
This paper reports the high antimicrobial activity of the copper-graphene-ZnO composite. The topic is interesting and is within the scope of the journal. In general, I recommend that the article needs minor revisions.
The abstract line: "Thus, this nanocomposite could be an effective alternative to controlling food- and drug-resistant pathogens gram positive and gram negative bacterial pathogens---need to be refined.
Material methods lack the docking methods that need to be incorporated.
In the case of bacteria, it should be italic, which needs to be checked throughout the manuscript.
Mark the graphene, Cu, and ZnO in the morphological characterizations.
The SEM scale should be more clear.
Why was the inhibition in rGO@ZnO-Cu stronger against gram-positive bacteria as compared to gram negative bacteria?
The abbreviations should be consistent throughout the manuscript.
The typo errors need to be corrected.
Author Response
Dear Reviewer
Thank you so much for your nice comment that will help to improve the soundness and novelty of this paper. The corrections and suggestions have been incorporated accordingly.
Comment-1 The abstract line: "Thus, this nanocomposite could be an effective alternative to controlling food- and drug-resistant pathogens gram positive and gram negative bacterial pathogens---need to be refined.
Response:1 The comments and suggestions have been incorporated
Comment:2 Material methods lack the docking methods that need to be incorporated.
Response:2 The material methods related to docking have been incorporated
Comment: 3 In the case of bacteria, it should be italic, which needs to be checked throughout the manuscript.
Response:3 The manuscript have been checked and required correction have been incorporated
Comment:4 Mark the graphene, Cu, and ZnO in the morphological characterizations.
Response:4 The comments and suggestions have been incorporated
Comment:5 The SEM scale should be more clear.
Response:5 The comments and suggestions have been incorporated
Comment:6 Why was the inhibition in rGO@ZnO-Cu stronger against gram-positive bacteria as compared to gram negative bacteria?
Response:6 The comments and suggestions have been incorporated
Comment:7 The abbreviations should be consistent throughout the manuscript.
Response:7 The manuscript have been checked and required correction have been incorporated
Comment:8 The typo errors need to be corrected.
Response:8 The manuscript have been checked and required correction have been incorporated.
Reviewer 2 Report
Ahmad and Ansari tried to synthesize the nano complex of Copper-Graphene-ZnO, and perform different characterisation test to check the appraisal of nano complex. I like the work and it is of highest importance. The is written well but I have some suggestion maybe those will help to improve the soundness and novality of this paper.
1. Need to improve abstract section as it is the first point for reader to know about the data provided.
2. Need to improve the rationale of the study in the introduction section. Prefer to add more relevant literature and discuss the results to draw the conclusion for the current hypothesis.
3. If possible add a diagram regarding the synthesis process of nano complex
4. Add description in characterization heading and mention some detail about each characterization perform.
5. Mention the principle for all charterization techniques
6. Need to add results for particle size analysis, zeta potential as these are important parameters to know about prepared nanocomplexes were of nano size range.
7. Need to mention the software name from which docking studies were performed
8. Need to improve the discussion section in general.
Author Response
Dear Reviewer
Thank you so much for your nice comment that will help to improve the soundness and novelty of this paper. The corrections and suggestions have been incorporated and highlighted accordingly.
Comment 1: Need to improve abstract section as it is the first point for reader to know about the data provided.
Response-The abstract has been revised and improved.
Comment 2. Need to improve the rationale of the study in the introduction section. Prefer to add more relevant literature and discuss the results to draw the conclusion for the current hypothesis.
Response-The introduction has improved with citing ref as per reviewer suggestions.
Comment 3. If possible add a diagram regarding the synthesis process of nano complex.
Response 3:A diagram showing synthesis process has been added.
Comment 4. Add description in characterization heading and mention some detail about each characterization perform.
Response-4 The comments and suggestions have been incorporated.
Comment 5. Mention the principle for all charterization techniques.
Response-5 The comments and suggestions have been incorporated.
Comment 6. Need to add results for particle size analysis, zeta potential as these are important parameters to know about prepared nanocomplexes were of nano size range.
Response-6 The SEM/and TEM results have been described.
Comment 7. Need to mention the software name from which docking studies were performed.
Response- The comments and suggestions have been incorporated.
Comment 8. Need to improve the discussion section in general.
Response-8 The Discussion section has been improved.
Round 2
Reviewer 1 Report
The authors have adequately addressed the raised concerns.
Author Response
Dear Reviewer
Thanks for your comments and suggestions.
We've taken all of the comments into account, and the editing service was used to improve the English.

Reviewer 2 Report
Sorry for late review, I am out of office and now I checked authors extensively revised the manuscript and alot of my comments authors resolved already. Now my decision is accept manuscript in current form.
Author Response

(The authors gave the same response as above.)
